# Stock prediction based on bidirectional gated recurrent unit with convolutional neural network and feature selection

**Qihang Zhou[1], Changjun Zhou**  **[1]\*, Xiao Wang[2]**

**1** College of Mathematics and Computer Science, Zhejiang Normal University, Jinhua, China, **2** Xingzhi College Zhejiang Normal University, Jinhua, China

\* zhou-chang231@163.com

## Abstract

With the development of recent years, the field of deep learning has made great progress. Compared with the traditional machine learning algorithm, deep learning can better find the rules in the data and achieve better fitting effect. In this paper, we propose a hybrid stock forecasting model based on Feature Selection, Convolutional Neural Network and Bidirectional Gated Recurrent Unit (FS-CNN-BGRU). Feature Selection (FS) can select the data with better performance for the results as the input data after data normalization. Convolutional Neural Network (CNN) is responsible for feature extraction. It can extract the local features of the data, pay attention to more local information, and reduce the amount of calculation. The Bidirectional Gated Recurrent Unit (BGRU) can process the data with time series, so that it can have better performance for the data with time series attributes. In the experiment, we used single CNN, LSTM and GRU models and mixed models CNN-LSTM, CNN-GRU and FS-CNN-BGRU (the model used in this manuscript). The results show that the performance of the hybrid model (FS-CNN-BGRU) is better than other single models, which has a certain reference value.

## Introduction

With the development of China's economy and the improvement of people's living standards, the stock market has become a hot area of attention. Although China's stock market started late, its growth momentum can't be underestimated. Therefore, scholars and investors in various countries are very optimistic about the future development trend of China's stock market. In the stock market, there are thousands of stocks in China alone, and each stock is affected by many factors. Therefore, it is particularly difficult to select one or several high quality stocks from many stocks. The change of stock price is influenced by many factors, such as natural disasters, the influence of politicians and the influence of the country in the world. Because the change of stock price is nonlinear, it is very important for many researchers and investors [1] to predict the trend of stock price in advance. The establishment of a high precision and reasonable stock prediction model can effectively reduce the loss of investors in the stock market, but also can improve their control of the stock price. Stock is a kind of non-linear data with

**Funding:** This work is supported by the National Natural Science Foundation of China (Nos.61672121).

**Competing interests:** The authors have declared that no competing interests exist.

time series attribute, so [2] proposed a cumulative autoregressive moving average method combined with least squares support vector machine (ARI-MA-LS-SVM) for stock prediction. The results show that the model has good universality and stability, and can provide a certain reference value for many investors and research institutes. [3] proposed a feature selection algorithm based on weighted sum least squares support vector machine (LS-SVM). We first add analytic hierarchy process (AHP) to the stock, and give the corresponding evaluation index. The features obtained by AHP method are added to LS-SVM model and the conclusion is drawn. The results show that the performance of this model is better than other models.

In the 21st century, the rapid development of science and technology has also led to the development of deep learning, so there are many researches on deep learning [4, 5], such as face recognition [6], emotion recognition [7] and image recognition [8]. Compared with other traditional machine learning algorithms, deep learning can deal with nonlinear problems well, so deep learning can play an important role in many research fields. [9] use a hybrid model based on Convolutional Neural Network (CNN) and support vector machine (SVM) to predict the stock index. The results show that the neural network can deal with continuous and classi-fied forecasting variables. [10] adopts a hybrid model (RNN + LSTM)and the results show that the hybrid model has a good application prospect for the stock price forecast of single stock with variables such as corporate behavior and corporate announcement. [11] proposed a new appearance model, which can be embedded into the recurrent neural network of bidirectional short-term memory unit and can effectively learn to track. It is superior to most other models in the benchmark video. [12] proposed a hybrid model based on neural network and B-P algo-rithm to predict stock price. The results show that the accuracy rate of using a single fuzzy algo-rithm is 62.12%, while the accuracy rate of using a single B-P algorithm is 73.29%, and the effect is the best. After comparing the influence of different hidden layers of neural network on the results, it is found that the accuracy rate of B-P neural network is better than that of fuzzy algorithm, and the algorithm provides a certain reference for stock investors The value of examination. Taking block chain information as the main research object, [13] discusses the impact of official information on the stock price trend caused by investors' intervention in stocks, and predicts stocks according to investors' preferences, and puts forward a stock predic-tion model based on LSTM. The results show that the prediction ability of the model is improved after adding the emotional characteristics, which reflects that information clarifica-tion is helpful to the prediction of stock price. [14] proposed a hybrid model based on inte-grated EMD and LSTM. The complex stock data is decomposed into smoother subsequences than the original time series by using the comprehensive dynamic decomposition method, and then the decomposed data is put into LSTM model for training and prediction, and compared with other five prediction methods. The results show that the hybrid model proposed in this paper has higher prediction accuracy than other models. [15] proposed a hybrid stock forecast-ing model based on CNN-LSTM, and add MLP, CNN, RNN, LSTM and CNN-LSTM to make horizontal comparison. The results show that the accuracy of the CNN-LSTM stock forecasting model is higher than other forecasting models. [16] propose a stock prediction model based on LSTM and attention mechanism. Firstly, wavelet transform is used to denoise the stock data, and then S P500, DJIA, HSI stock data are put into the prediction model for calculation. The results show that the stock prediction model proposed in this paper has the best effect.

## Research methodology

### Feature selection (fs)

Feature selection [17] is a very important part in the process of data preprocessing. To a certain extent, it can select important features in order to increase the training speed and alleviate the

dimension disaster. At the same time, it can increase the accuracy of the training results by removing the inconsistent features. [18] take the glass bottle defect detection as a classification problem, and extract features from the glass bottle knock signal as the data input. In this paper, an improved feature selection algorithm (SFLA-ImRMR-BP) is proposed based on the previous minimum redundancy maximum correlation algorithm (SFLA-ImRMR). The results show that the algorithm proposed in this paper has a great improvement in accuracy compared with previous algorithms. [19] proposed an algorithm based on DNP-AAP (deep neural pursuit average activation potential). The results show that the algorithm can effectively identify the known AMR related genes, and it also provides a list of candidate genes that may lead to the discovery of new AMR factors, which provides a new scheme for microbiologists. The most representative feature selection methods are as follows:

**Filter selection.** Relief (relative features) is a well-known filtering feature selection method, which assumes that the importance of feature subset is determined by the sum of feature values corresponding to each subset in each subset. The calculation formula is as follows:

$$\delta^j = \sum_i - \text{diff}\left(x_i^j, x_{i,mh}^j\right)^2 + \text{diff}\left(x_i^j, x_{i,nm}^j\right)^2, \tag{1}$$

Where $x_i^j$ is the value of sample $x_i$ on sample $j$. If the attributes are discrete, then *diff* is 0 if and only if the attributes are equal, and 1 for all other times. If it is continuous, then *diff* is the distance, and the larger the value, the better.

The construction method is as follows: Selecting the nearest neighbor $x_{i,nh}$ from the similar samples of $x_i$, which is called guessing the nearest neighbor. Selecting a nearest neighbor $x_{i,nm}$ in heterogeneous samples is called guessing the wrong nearest neighbor.

It can be seen from the above formula that for the value corresponding to a feature, the closer the same kind is, the farther the different kind is, and the larger the corresponding statistics will be.

**Embedded selection.** When the dimension of training data is large and the training data is small, it is easy to over fit, so it is necessary to add a regular term. L1 norm is easier to get sparse solutions than L2 norm, so the learning method based on L1 regularization is an embedded feature selection method. The Proximal gradient descent (PGD) [20] method can be used to solve the L1 regularization problem. PGD can make LASSO and other methods based on L1 norm minimization be solved quickly.

## Convolutional neural network (cnn)

Convolutional neural network [21–23] is a kind of feed forward neural network with convolution calculation and depth structure, which is one of the most representative algorithms of deep learning. Convolutional neural network has the ability of representation learning. It can extract the features of input data according to its hierarchical structure, and select the useful information in the data. At the same time, it also reduces a lot of calculation, which greatly reduces the training time. The research of convolutional neural network began in the 1980s and 1990s. Convolutional neural network consists of the following parts: convolution layer, pooling layer, activation function and output layer. Each convolution layer contains many convolution cores. Although the features of the data are greatly extracted after convolution operation, a common fault of convolution calculation is that the problem of high dimension appears. Therefore, adding pooling layer after convolution calculation can effectively solve the problem of high data dimension, improve the robustness of the extracted features, and then put the data into the activation function to fit nonlinear problems. The specific calculation

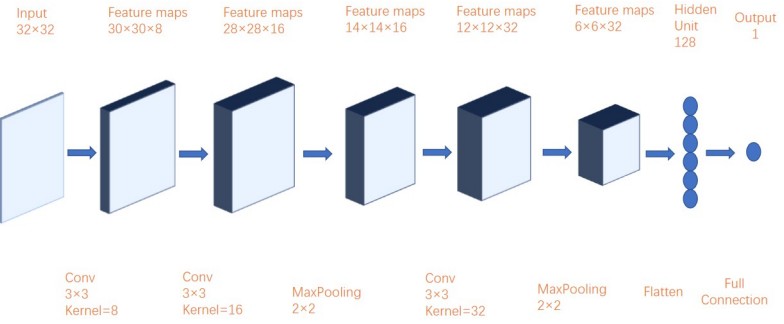

**Fig 1. Convolution process.**

process is shown in formula (2), and the convolution image is shown in Fig 1.

$$P_t = \text{Relu}(x_t * w_t + b_t), \tag{2}$$

Where $P_t$ is the output result, Relu is the activation function, $x_t$ is the input data, $w_t$ is the weight of convolution kernel, and $b_t$ is the bias.

## Bidirectional gated recurrent unit (bgru)

RNN [24–27] is a recurrent neural network, which takes sequence data as input, recursively along the evolution direction of the sequence, and all nodes are connected by chain. The research on Recurrent Neural Network began in the 1980s and 1990s, and developed into one of the deep learning algorithms at the beginning of the 21st century. Bidirectional Recurrent Neural Network and Long Short-Term Memory Network [16, 28–30] are common Recurrent Neural Networks. but they have the problems of gradient disappearance and gradient explosion. LSTM is an improved version based on RNN, which can solve the problem of gradient vanishing and gradient explosion. It has a good performance in the fields of speech recognition, language modeling and machine translation. At the same time, it is also used in all kinds of problems with time series attributes. To understand Bidirectional GRU, we must first understand GRU [31, 32]. GRU is a variant of LSTM network, which is simpler than LSTM. Unlike LSTM, which has three gates (input gate, forgetting gate and output gate [33]), GRU has only two gates (update gate and reset gate [34]). The function of update gate is similar to forgetting gate and input gate in LSTM. It determines which information to forget and which new information to add and update. The reset gate is used to decide which part of the previous information is not important for the current time calculation. Because the number of gating of GRU is less than LSTM, GRU is faster than LSTM in calculation. The structure of GRU is shown in Fig 2. GRU calculation formula is as follows:

$$z_t = \sigma \left( W_z \cdot [h_{t-1}, x_t] \right), \tag{3}$$

$$r_t = \sigma \left( W_r \cdot [h_{t-1}, x_t] \right), \tag{4}$$

$$h_t = (1 - z_t)^* h_{t-1} + z_t * \tilde{h}_t, \tag{5}$$

$$\tilde{\mathbf{h}}_t = \tanh \left( W \cdot [r_t^* h_{t-1}, x_t] \right), \tag{6}$$

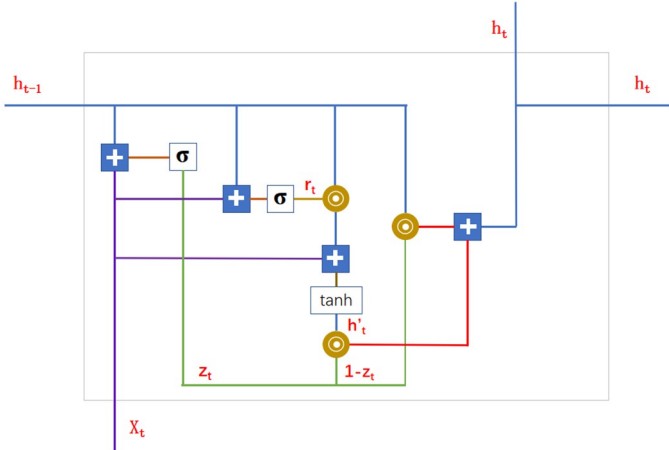

**Fig 2. GRU structure diagram.**

$$S(x) = \frac{1}{1 + e^{-x}}, \tag{7}$$

$$S'(x) = \frac{e^{-x}}{(1 + e^{-x})^2} = S(x)(1 - S(x)), \tag{8}$$

Where $\sigma$ is the sigmoid activation function, which is used to compress all the output data between 0 and 1. The sigmoid function and its derivation formula are shown in formula 7 and formula 8. $W_z$ and $W_r$ are the weights of update gate $z_t$ and reset gate $r_t$ respectively. $h_{t-1}$ is the past information, $h_t$ contains the past information $h_{t-1}$ and the present information $\tilde{h}_t$. The present information is determined by the past information $h_{t-1}$ and the current input. The model structure of Bidirectional GRU [35–37] is similar to GRU model. There is a positive time series and a reverse time series. The results corresponding to the last state of the positive time series and the reverse time series are combined as the final output results. The model can make use of the past and future information at the same time. In this paper, we use Bidirectional GRU model. The network in Fig 3 contains two sub networks: forward status and backward status, which represent forward transmission and backward transmission respectively.

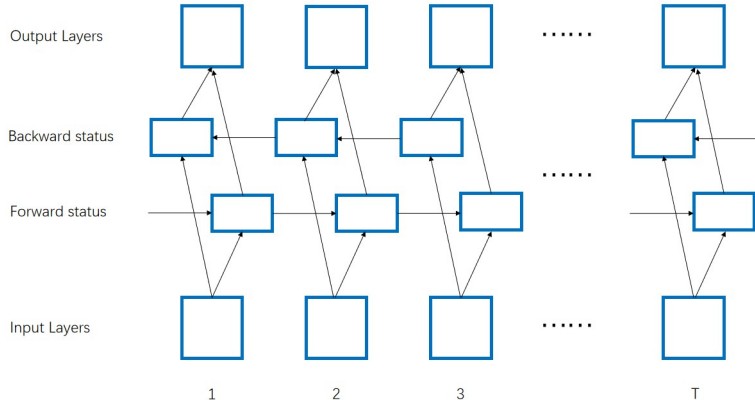

**Fig 3. BGRU structure diagram.**

## Method based on fs-cnn-bgru

The hybrid model based on FS-CNN-BGRU proposed in this paper includes FS, CNN and BGRU. FS is used for feature selection, CNN is used for feature extraction, and BGRU is used for processing data with time series. In this paper, the hybrid model is divided into six parts: input layer, FS Normalization layer, CNN layer, BGRU layer, full connection layer and output layer. Most of the prediction methods used are CNN-BGRU. The CNN module includes convolution kernel, pooling layer and flatten layer. Convolution layer can be set to multilayer, so it has better ability of feature extraction. Similarly, setting the convolution core size can also improve the ability of feature selection. In order to maximize the utilization of data, the size of convolution kernel is usually set to n × n. The pool layer can also be set with multiple layers, and the size of each pool layer can also be set to n × n. Then the extracted data of CNN feature is sent to BGRU layer. Because BGRU has the ability to process time series attributes, the prediction accuracy of the model can be effectively improved by increasing the number of layers of BGRU and the number of units of each layer of BGRU. At the same time, the occurrence of over fitting can be well prevented by setting the dropout layer. Data can be output through BGRU layer, full connection layer and output layer. The FS-CNN-BGRU hybrid model proposed in this paper is improved based on CNN-GRU model. The practicability of CNN-GRU model has been elaborated in many papers, and good results have been achieved. FS-CNN-BGRU model uses the features extracted from CNN to be put into BGRU for prediction. BGRU has the ability to deal with time series very well. In time series, BGRU can solve the problem of gradient disappearance and gradient descent, and it has better performance than the GRU. Then in general, the stock data is many and miscellaneous, there are a lot of data for the model performance improvement is not big or has a reaction, so we need to use FS method to select features, so that the model can play the best performance. This paper constructs a complete process based on FS-CNN-BGRU model. The specific experimental flow is shown in the "Experiment".

If the prediction result is not good, adjust the number of CNN convolution cores and the number of BGRU units to get better results. The process of this experiment is shown in Fig 4. Firstly, the data is acquired, and then the acquired data is put into the FS-N layer (Feature Selection and Data Normalization), and then the data can get the data that can well represent the high dimensional features of the original data after passing through the CNN layer, and then the data is put into the bidirectional GRU with time series attributes, and finally the

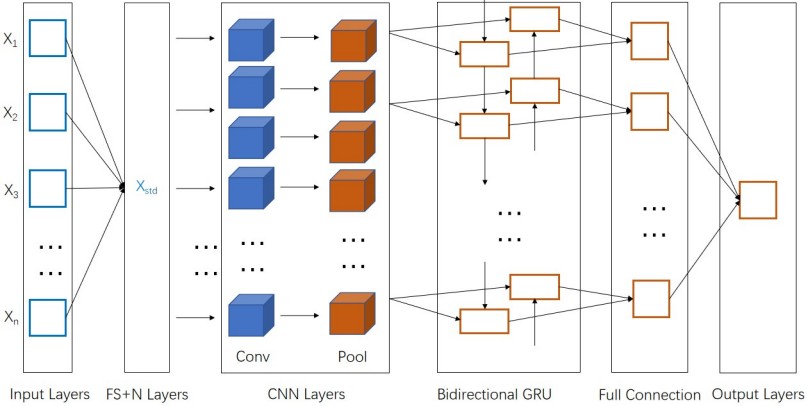

**Fig 4. Experiment flow chart.**

results are obtained and predicted through the full connection layer and output layer to evaluate the performance of the model.

## Experiment

### Get experimental data

Through various ways to obtain data, we select 8 dimensional data of Shanghai Composite Index, Shenzhen Composite index, CSI 300, Growth Enterprise Index, China National Petroleum Corporation (CNPC), China State Construction Engineering Corporation (CSCEC), China Railway Rolling stock Corporation (CRRC) and Shanghai Automotive Industry Corporation(SAIC) as input data, including the opening price, the highest price, the lowest price, the previous closing price and trading volume, and the closing price as forecast data. Some data of Shenzhen Composite Index is shown in Table 1.

### Data preprocessing

In order to induce the statistical distribution of sample data and improve the training speed of neural network, we need to normalize the data. The normalization formula is as follows:

$$X_s caled = \frac{(X - X \cdot \min(axis = 0))}{(X \cdot \max(axis = 0) - X \cdot \min(axis = 0))} \cdot (\max - \min) + \min, \qquad (9)$$

Where $X \cdot \min(axis = 0)$ is the row vector composed of the minimum value in each column, and $X \cdot \max(axis = 0)$ is the row vector composed of the maximum value in each column. and are the maximum and minimum values respectively, and their default values are 1 and 0 respectively. $X_{std}$ and $X_{scaled}$ are the results of standardization and normalization respectively.

### Data division

This paper uses the opening price, the highest price, the lowest price, the previous closing price, the range of rise and fall, the amount of rise and fall and the trading volume of Shanghai Composite Index, Shenzhen Composite Index, CSI 300, Growth Enterprise Index, CNPC, CSCEC, CRRC and SAIC from January 1991 to December 2020as the data input and the closing price as the data output. At the same time, the proportion of the data is divided into 8:2 That is, the first 80% of the data is used as the training set, and the last 20% as the test set.

### Experiment environment

This experiment uses Intel i7-10700 8 core 16 thread processor, 16GB memory, win10 operating system, anaconda3 as the experimental platform, python language programming. The deep learning framework uses Keras in Tensorflow 2.0, which is powerful and concise. It provides users with many interfaces to help them quickly build their own models.

**Table 1. Some data of Shenzhen Composite Index.**

| No | Closing | Hight | Low | Open | Previous Closing | Rise and Fall | Up and Down | Volume |
|----|---------|-------|-----|------|------------------|---------------|-------------|--------|
| 1 | 14134.9 | 14134.9 | 13819.7 | 13858.7 | 13854.1 | 280.7 | 2.0 | 1.47E+11 |
| 2 | 13854.1 | 13939.9 | 13806.6 | 13911.8 | 13889.9 | -35.8 | -0.3 | 1.22E+11 |
| 3 | 13889.9 | 13901.8 | 13688.8 | 13746.3 | 13751.1 | 138.8 | 1.0 | 1.30E+11 |
| 4 | 13751.1 | 13807.2 | 13709.8 | 13791.4 | 13763.3 | -12.2 | -0.1 | 1.11E+11 |
| 5 | 13763.3 | 13783.0 | 13641.7 | 13682.3 | 13692.1 | 71.2 | 0.5 | 1.08E+11 |

## The construction of model

The normalized data set is divided into training set and test set, and then the training set is put into the model for training. However, as the input value of the neural network before the non-linear transformation increases with the depth of the network, its distribution will gradually shift or change, which leads to the gradual disappearance of the gradient of the underlying neurons in the back propagation, which is why the convergence speed of the neural network will be slower and slower. Batch normalization (BN) [38] is to force the input number of neurons in any layer back to the normal distribution with the mean value of 0 and the variance of 1 by some normalization methods, which can solve this kind of problem well and greatly accelerate the convergence speed of neural network in the training process. The calculation formula of BN is as follows:

$$\mu_{\mathcal{B}} \leftarrow \frac{1}{m}\sum_{i=1}^{m} x_i, \tag{10}$$

$$\sigma_{\mathcal{B}}^2 \leftarrow \frac{1}{m}\sum_{i=1}^{m} (x_i - \mu_{\mathcal{B}})^2, \tag{11}$$

$$\hat{x}_i \leftarrow \frac{x_i - \mu_{\mathcal{B}}}{\sqrt{\sigma_{\mathcal{B}}^2 + \epsilon}}, \tag{12}$$

$$y_i \leftarrow \gamma\hat{x}_i + \beta \equiv \mathrm{BN}_{\gamma,\beta}(x_i), \tag{13}$$

Where $x_i$ is the input value, $y_i$ is the output value after BN normalization, and m is the size of each mini batch, that is, the mini batch with m inputs. $\mu_{\mathcal{B}}$ is the average of all inputs in the same Mini batch. $\sigma_{\mathcal{B}}^2$ is the variance of all outputs in the same Mini batch. Then the normalized value is obtained by $\mu_{\mathcal{B}}$ and $\sigma_{\mathcal{B}}^2$, and the formula (12) is brought into the formula (13) to obtain the output value $y_i$, $\gamma$ and $\beta$ are obtained by machine learning. So far, the BN algorithm can be used to normalize the data of neurons in each layer of neural network, so as to improve the training speed of neural network.

In BGRU layer, MSE is used as loss function, Adam as optimizer, learning rate is 0.001, batch size is 64. The purpose of training is to make the predicted value and the real value as small as possible. In order to ensure that the experimental data are real and effective, we only put the training data into the model for training, and the test set does not participate in the training. Each training data is put into the FS-CNN-BGRU model to calculate the predicted value, and the difference between the real value and the predicted value is compared. Then the gradient descent algorithm in the optimizer is used to update the weight of each parameter in the FS-CNN-BGRU model. With the continuous updating and iteration, the prediction results of FS-CNN-BGRU model will be more and more accurate. After the training of FS-CNN-BGRU model, put the test set into the model to predict and get the prediction results, and then compare the prediction results with the real value to get the error, so as to evaluate the performance of FS-CNN-BGRU stock prediction model. However, there are many problems in the process of model training. For example, underfitting and overfitting, underfitting is caused by insufficient data or insufficient training times. The solution is very simple, that is to increase the amount of data and training times. The main performance of overfitting is that it performs well in the training set, but it does not perform well in the test integration. There are two ways to solve this problem: (1) train the results before the overfitting occurs, so as to obtain the best performance. (2) Increase the proportion of dropout layer and dropout.

Therefore, adding dropout appropriately in CNN and BGRU layers can effectively suppress the occurrence of overfitting.

## Evaluating indicator

The evaluation indexes used in this paper are MAPE and R2. The two evaluation indexes are shown in formulas 14 and 15.

$$MAPE = \frac{1}{n}\sum_{i=1}^{n}\frac{|\hat{y}^{(i)} - y^{(i)}|}{y^{(i)}} \times 100\%, \tag{14}$$

Where n is the number of samples, $\hat{y}^{(i)}$ is the predicted value, and $y^{(i)}$ is the real value.

$$R^2 = \frac{\sum_{i}^{m}(\widehat{y_i} - y_i)^2}{\sum_{i}^{m}(\widehat{y_i} - \bar{y_i})^2}, \tag{15}$$

Where m is the number of samples, $\hat{y}_i$ is the predicted value, $y_i$ is the real value, and $\bar{y}_i$ is the sample mean value.

## Experimental result

In order to improve the persuasiveness of FS-CNN-BGRU model results, CNN, LSTM, GRU, CNN-LSTM and CNN-GRU methods are added for horizontal comparison, and all methods used in the test use the same number of cores and units to eliminate the influence of all different factors.

Firstly, we divide the eight stocks mentioned above into two categories: index stocks and common stocks, and then select the data from January 1, 1991 to December 31, 2020 as the data input, and then, the number of convolution kernels is 8, and the data not using feature selection and using feature selection are put into the model to view the results. Table 2 shows the results of the selection of non use feature selection and use feature selection of index stocks (%).

It can be seen from Table 2 that the MAPE values of four index stocks after using feature selection are lower than those without using feature selection. In order to further verify the effectiveness of feature selection, four common stocks are also compared by using feature selection. Table 3 shows the results of common stock not using feature selection and using feature selection (%).

From Table 3, it can be concluded that the MAPE values of four common stocks after using feature selection are lower than those without using feature selection, so it can be concluded

**Table 2. MAPE value of index stock obtained by different methods (%).**

| Method/Stock | Shenzhen Composite Index | CSI 300 | Shanghai Composite Index | Growth Enterprise Index |
|---|---|---|---|---|
| Common | 6.5435 | 2.2448 | 3.7110 | 3.0161 |
| Feature Selection | **5.2200** | **1.8972** | **1.4878** | **2.5249** |

**Table 3. MAPE value of common stock obtained by different methods (%).**

| Method/Stock | CNPC | CSCEC | CRRC | SAIC |
|---|---|---|---|---|
| Common | 4.2740 | 2.3357 | 3.1741 | 6.9912 |
| Feature Selection | **2.7476** | **1.8167** | **2.4024** | **6.0518** |

**Table 4. MAPE values of index stocks in convolution kernels with different numbers (%).**

| Filters/Stock | Shenzhen Composite Index | CSI 300 | Shanghai Composite Index | Growth Enterprise Index |
|---|---|---|---|---|
| 8 | 5.2200 | 1.8972 | 1.4878 | 2.5249 |
| 16 | 4.3932 | 1.4162 | 1.3944 | 2.3679 |
| 32 | **2.0601** | **1.2630** | **1.1426** | **2.0635** |
| 64 | 2.4203 | 1.2754 | 1.2538 | 2.4133 |

that the MAPE values of two kinds of stocks after using feature selection are reduced, which meets the experimental requirements.

## Training model

**Determine the number of convolution kernels.** The next step is to test the influence of each number of convolution kernel and unit number on the final result (all the next experiments are the experimental data after adding feature selection), select 8, 16, 32 and 64 convolution cores in the convolution layer to compare their MAPE values. Table 4 shows the influence of different convolution cores in CNN layer on index stocks (%).

From Table 4, we can see that the error of the four index stocks decreases with the increase of the number of convolution kernels. When the number of convolution kernels in CNN layer is 32, the error reaches the minimum, but when the number of convolution kernels reaches 64, the error increases instead, which may be caused by overfitting.

Next, we test the performance of different convolution kernels in common stock. Table 5 shows the MAPE values (%) of different convolution kernels in common stock.

From Table 5, it is not difficult to see that in common stocks, with the increase of the number of convolution kernels, the accuracy of the model also decreases as the result of the previous table. At the same time, when the number of convolution kernels reaches 64, the error increases instead of decreasing, which confirms the conjecture in the previous table: with the increase of the number of convolution kernels, the error increases because model is overfitted.

**Determine the number of lstm units.** Next, we test the impact of different number of units on stock data in LSTM. Similarly, we also divide the data set into two categories: index stocks and common stocks, and test the results of different number of units on the performance of two types of stocks. The number of units selected in LSTM is the same as that of convolution kernel selected in the last experiment, which is 8, 16, 32 and 64 units. Table 6 shows the influence of different number of units on Index Stocks (%).

From Table 6, we can see that with the increase of the number of units, the error of each index stock also decreases. When the number of units reaches 64, the error is the smallest, which is lower than that of other different units. Then test the performance of different unit numbers in common stock. Table 7 shows the effect of different number of units on common stock (%).

**Table 5. MAPE value of common stock in convolution kernel with different number (%).**

| Filters/Stock | CNPC | CSCEC | CRRC | SAIC |
|---|---|---|---|---|
| 8 | 2.7476 | 1.8167 | 2.4024 | 6.0518 |
| 16 | 2.4746 | 1.5089 | 2.3298 | 5.9222 |
| 32 | **2.2877** | **1.4477** | **2.0507** | **4.4605** |
| 64 | 2.3147 | 1.4635 | 2.1675 | 5.0786 |

**Table 6. MAPE value of index stocks in different number of units (%).**

| Units/Stock | Shenzhen Composite Index | CSI 300 | Shanghai Composite Index | Growth Enterprise Index |
|---|---|---|---|---|
| 8 | 3.6526 | 1.3408 | 1.2452 | 2.0463 |
| 16 | 3.5708 | 1.2697 | 1.1991 | 1.8528 |
| 32 | 2.2845 | 1.2530 | 1.1397 | 1.8242 |
| 64 | **1.8654** | **1.1819** | **1.1049** | **1.7784** |

**Table 7. MAPE value of common stock in different number of units (%).**

| Units/Stock | CNPC | CSCEC | CRRC | SAIC |
|---|---|---|---|---|
| 8 | 2.8631 | 1.4263 | 2.0908 | 3.0975 |
| 16 | 2.6367 | 1.2880 | 1.9241 | 2.6892 |
| 32 | 2.4953 | 1.2587 | 1.8071 | 2.3812 |
| 64 | **2.3027** | **1.2461** | **1.7875** | **2.3514** |

From Table 7, we can see that with the increase of the number of units, the prediction accuracy of common stocks becomes higher and higher. When the number of units reaches 64, the error is the smallest, which is also consistent with the prediction of index stocks in Table 6. Because this paper is based on the hybrid model of FS-CNN-BGRU, after obtaining the best number of CNN and LSTM units, the number of GRU and BGRU units is set to 64, so in the next experiment, the number of convolution cores of CNN is finally selected as 32, and the number of units in BGRU is selected as 64 to ensure the correctness and unity of the experiment.

Table 8 shows the parameters of FS-CNN-BGRU model. The model is divided into four layers: input layer, convolution layer, BGRU layer and output layer. The convolution layer is set to 1 layer, the number of convolution cores is 32, the size of convolution core is $1 \times 1$, stripe is set to 1, padding is the same, Tanh is used as the activation function, Max-Pooling is used as the Pooling layer, padding is set to 1, and Relu is used as the activation function. The bidirectional GRU is set to 1 layer with 64 units. The BGRU network time step is set to 50 to predict the closing price of the 51st day. The gradient descent algorithm uses Adam and iterates 100 times.

**Table 8. Parameters of FS-CNN-BGRU model.**

| Parameter name | Parameter value |
|---|---|
| Network layers | 4 |
| Convolutional filters | 32 |
| Convolutional kernel size | $1 \times 1$ |
| Convolutional activation function | Tanh |
| Convolutional padding | Same |
| Pooling size | $1 \times 1$ |
| Pooling padding | Same |
| Pooling activate function | Relu |
| Number of BGRU layers | 64 |
| Batch size | 64 |
| Optimization | Adam |
| Epochs | 100 |

Table 9. MAPE values of different methods (%).

| Model | MAPE (%) |
|---|---|
| CNN | 2.0601 |
| LSTM | 1.8654 |
| GRU | 1.8332 |
| CNN-LSTM [14] | 1.6426 |
| CNN-GRU | 1.6354 |
| FS-CNN-BGRU | **1.4325** |

Taking the Shenzhen Composite Index as an example, we choose the data of the stock from January 1, 1991 to December 31, 2020 as the input data of the model, and add these data for normalization, and take the first 80% of the data as the training data, the last 20 as the test data, and use MAPE as the evaluation index of the model. In order to better test the performance of this model, we add five other methods for comparison. Through Feature Selection, we can calculate which features are helpful to improve the accuracy of the model, so as to achieve the compression of the dimension of feature space, that is, to obtain a group of data with less precision and least fitting error, CNN can well put forward the characteristics of the input data, and LSTM can have a good performance in the data with time series, so [14] proposed a CNN-LSTM stock prediction model.

Table 9 shows the MAPE values of different hybrid models in Shenzhen stock index. It is not difficult to see from Table 9 that the prediction error (MAPE) of FS-CNN-BGRU hybrid model is 1.4325%, that of CNN-GRU is 1.6354%, that of CNN-LSTM is 1.6426%, that of GRU is 1.8332%, that of LSTM is 1.8654%, and that of CNN is 2.0601%. From these data, the MAPE value of FS-CNN-BGRU is lower than that of CNN-GRU and other four models. It can be seen from Table 10 that the performance of FS-CNN-BGRU stock prediction model R2 proposed in this paper is also higher than other models. Although the performance of FS-CNN-BGRU hybrid model is higher than other models, there is still room for improvement.

**Comparison of results of various models.** In order to better verify the prediction ability of FS-CNN-BGRU model, seven other stocks are added for experimental comparison. Tables 11 and 12 show the performance of MAPE in index stocks and common stocks respectively (%). Tables 13 and 14 show $R^2$'s performance in index stocks and common stocks respectively. The forecast chart (real price and prediction price) of the eight stocks are shown in Figs 5–12.

By comparing CNN, LSTM, GRU, CNN-LSTM and CNN-GRU, the performance of FS-CNN-BGRU hybrid model is generally better. It can be seen from Tables 11 and 12 that the performance of CNN model is the worst for both index stocks and common stocks. The performance of LSTM and GRU models is better than CNN model, but the results are still not

Table 10. $R^2$ of different methods.

| Model | $R^2$ |
|---|---|
| CNN | 0.971365 |
| LSTM | 0.977786 |
| GRU | 0.978228 |
| CNN-LSTM [14] | 0.981674 |
| CNN-GRU | 0.980416 |
| FS-CNN-BGRU | **0.983808** |

Table 11. MAPE value of index stocks (%).

| Model | Shenzhen Composite Index | Growth Enterprise Index | CSI 300 | Shanghai Composite Index |
|---|---|---|---|---|
| CNN | 2.0601 | 2.0635 | 1.2630 | 1.1426 |
| LSTM | 1.8654 | 1.7784 | 1.1819 | 1.1049 |
| GRU | 1.8332 | 1.7675 | 1.0580 | 1.0877 |
| CNN-LSTM [14] | 1.6426 | 1.7641 | 1.0430 | 1.0139 |
| CNN-GRU | 1.6354 | 1.7540 | 1.0424 | 1.0749 |
| FS-CNN-BGRU | **1.4325** | **1.6684** | **0.9892** | **1.0614** |

Table 12. MAPE value of common stock (%).

| Model | CNPC | CSCEC | CRRC | SAIC |
|---|---|---|---|---|
| CNN | 2.2887 | 1.4477 | 2.0507 | 3.0975 |
| LSTM | 2.3027 | 1.2461 | 1.7875 | 2.3514 |
| GRU | 2.0583 | 1.2220 | 1.7651 | 2.5513 |
| CNN-LSTM [14] | 2.0538 | 1.1891 | 1.6374 | 2.3029 |
| CNN-GRU | 2.1456 | 1.2022 | 1.6376 | 2.1789 |
| FS-CNN-BGRU | **1.7033** | **1.1714** | **1.6081** | **2.0926** |

Table 13. $R^2$ of index stocks.

| Model | Shenzhen Composite Index | Growth Enterprise Index | CSI 300 | Shanghai Composite Index |
|---|---|---|---|---|
| CNN | 0.971365 | 0.985536 | 0.977826 | 0.977393 |
| LSTM | 0.977786 | 0.990884 | 0.979814 | 0.978607 |
| GRU | 0.978228 | 0.990928 | 0.983899 | 0.979678 |
| CNN-LSTM [14] | 0.981674 | 0.990064 | 0.984244 | 0.981210 |
| CNN-GRU | 0.980416 | 0.989636 | 0.984310 | 0.979707 |
| FS-CNN-BGRU | **0.983808** | **0.991125** | **0.985729** | **0.979842** |

very ideal. From the perspective of index stocks, the forecast results of FS-CNN-BGRU hybrid model are 1.4325%, 1.6684%, 0.9892% and 1.0614% respectively, which are higher than other hybrid models. From the perspective of common stocks, FS-CNN-BGRU hybrid model's forecast results are 1.7033%, 1.1714%, 1.6081% and 2.0926%, which are also higher than other hybrid models. However, it is not difficult to find that the CNN-LSTM stock prediction model proposed in [14] outperforms CNN-GRU in the first three stocks. This may be because the gradient of CNN-LSTM hybrid model in the three stocks is deep and the error is small, so the

Table 14. $R^2$ of common stock.

| Model | CNPC | CSCEC | CRRC | SAIC |
|---|---|---|---|---|
| CNN | 0.993983 | 0.991586 | 0.965107 | 0.995745 |
| LSTM | 0.993494 | 0.994723 | 0.973520 | 0.997467 |
| GRU | 0.993616 | **0.995183** | 0.974131 | 0.997353 |
| CNN-LSTM [14] | 0.994447 | 0.994956 | 0.975138 | **0.997888** |
| CNN-GRU | 0.994152 | 0.995007 | 0.976248 | 0.997752 |
| FS-CNN-BGRU | **0.995694** | 0.995020 | **0.978322** | 0.997749 |

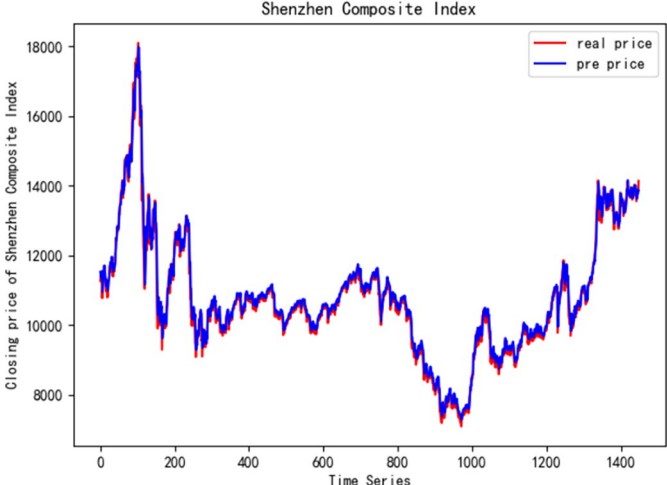

**Fig 5. Closing price of Shenzhen Composite Index.**

prediction result is higher than that of CNN-GRU hybrid model. However, the performance of FS-CNN-BGRU hybrid prediction model proposed in this paper is better than that of CNN-LSTM prediction model proposed in [14].

However, it can be seen from Tables 13 and 14 that the FS-CNN-BGRU model proposed in this paper performs well in CNPC and CRRC, but not well in the other two stocks, and the CNN-LSTM model performs best in SAIC. From Tables 11 and 14, we can find that the FS-CNN-LSTM stock forecasting model proposed in this paper has achieved the best results compared with other models, but from Tables 13 and 14, we can know that the model has room for optimization.

All the experimental data prove the accuracy and effectiveness of FS-CNN-BGRU hybrid model in stock forecasting. Although the prediction results of CNN, LSTM, GRU, CNN-LSTM

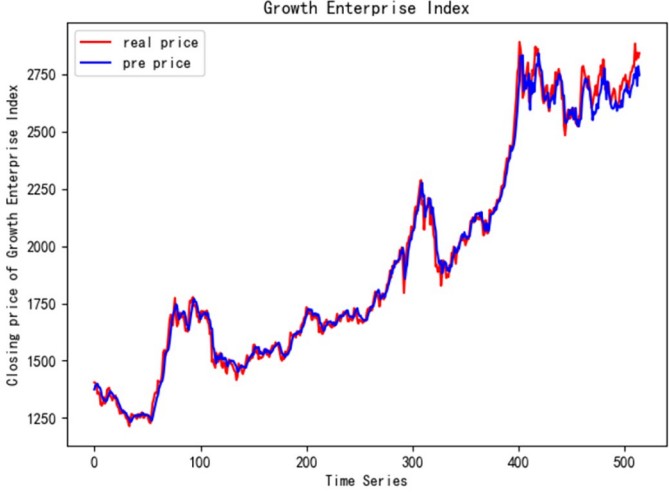

**Fig 6. Closing price of Growth Enterprise Index.**

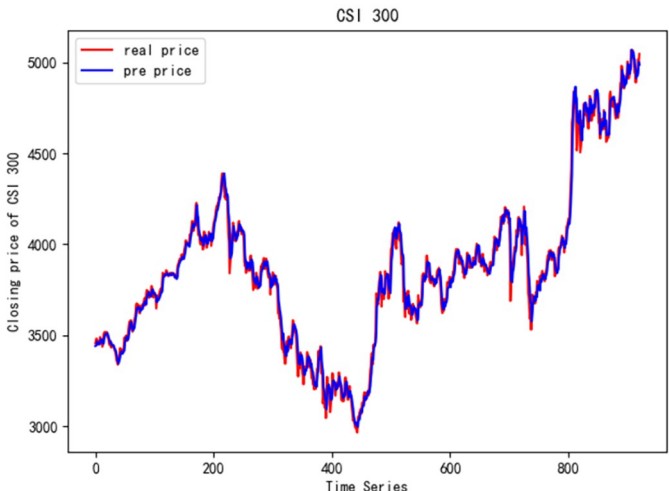

**Fig 7. Closing price of CSI 300.**

and CNN-GRU hybrid models are also very good, the prediction results of FS-CNN-BGRU are better than them to some extent. This further proves that it is feasible to use CNN to extract features and then use BGRU to predict the data with time series attributes. It provides a new investment idea for shareholders and stock investors.

## Conclusion

In this paper, convolutional neural network is responsible for feature extraction. It convolutes the features of the input data to obtain high-order features that can represent the data. LSTM and GRU can process the data with time series attributes because of their own unique attributes. Although a single neural network model (CNN, LSTM and GRU) can better predict the trend of stock closing price, the stock price is still affected by many factors, such as natural

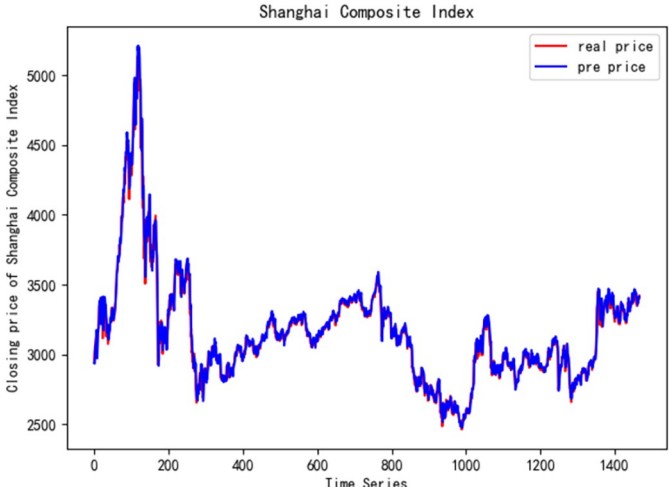

**Fig 8. Closing price of Shanghai Composite Index.**

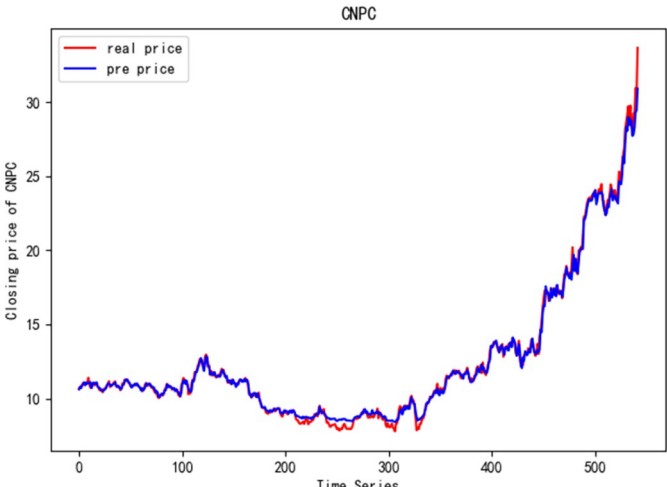

**Fig 9. Closing price of CNPC.**

disasters, people's emotions, politicians' attitudes and the management ability of the local government. Therefore, a single stock prediction model can't further predict the stock price, so there is an attempt to apply it The idea of merging multiple single models into a new hybrid model. According to the characteristics of stocks with many dimensions and huge and complex stock data, this paper proposes a stock forecasting model based on FS-CNN-GRU, which takes the data other than the closing price as the data input and the closing price as the output of the model, sets the time step to 50 and forecasts the closing price of the 51st day, and adds some other methods as horizontal comparison to judge the validity of the model Actual performance. The experimental results show that the proposed stock forecasting model has lower error than other methods, which proves the availability of the model in stock forecasting.

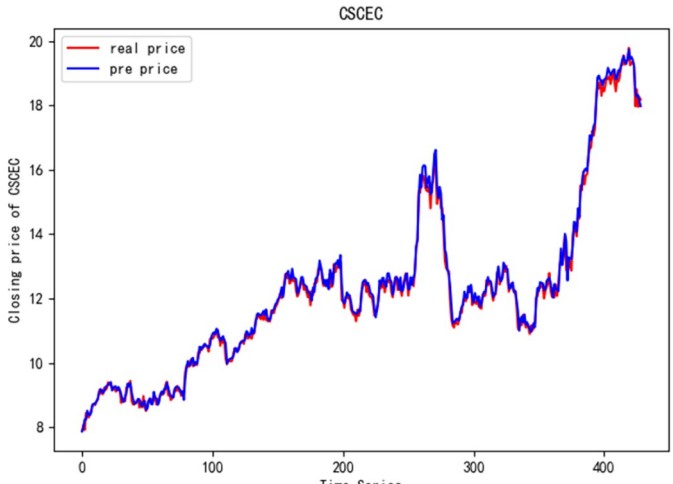

**Fig 10. Closing price of CSCEC.**

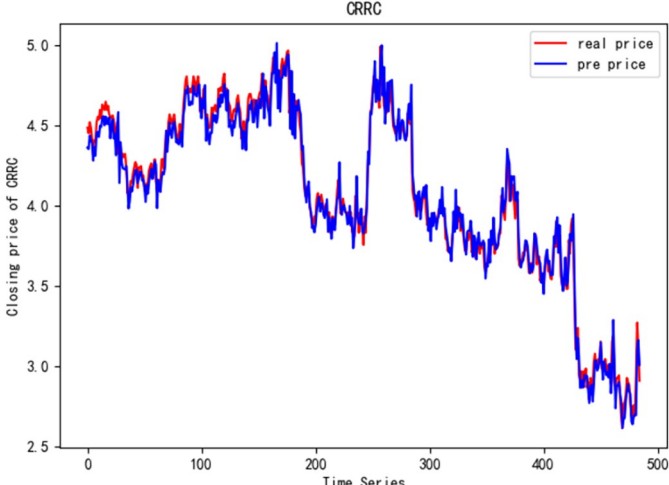

**Fig 11. Closing price of CRRC.**

Although the model proposed in this paper can achieve good performance in stock forecasting, it still faces many challenges in stock forecasting: first, the stock data used in this paper are all from China's stock market, and do not predict foreign stock markets. Second, the parameter adjustment of CNN and BGRU needs to be further strengthened. It is believed that the model can achieve better results among the stocks through in-depth parameter adjustment. In addition, although FS-CNN-BGRU hybrid model is used in this paper, I believe there are many hybrid models not mentioned in this paper that can further reduce the error of stock forecasting model, which is what we need to do, At the same time, some algorithms [39] can be added to the model to improve the effectiveness of the model. However, I think the FS-CNN-BGRU hybrid stock forecasting model proposed in this paper can help investors and investors make correct decisions in the stock market to a certain extent.

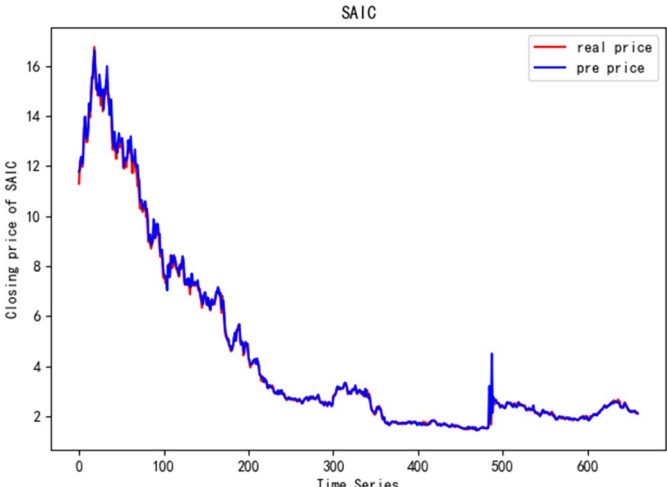

**Fig 12. Closing price of SAIC.**

## Acknowledgments

QIHANG ZHOU was born in Jiangshan, China in 1996. he received the bachelor's degree in Xingzhi College Zhejiang Normal University in 2019. He is currently pursuing a master's degree in Computer Science and Technology from Zhejiang Normal University, Jinhua, Zhejiang, CHN. His research interests include DNA coding design, machine learning and deep learning.

CHANGJUN ZHOU was born in Shangrao, China in 1977. He received Ph.D. degree in Mechanical Design and Theory from the School of Mechanical Engineering, Dalian University of Technology, Dalian in 2008. He is currently a professor at Zhejiang Normal University. His research interests include intelligence computing, pattern recognition, DNA computing. He has published 60 papers in these areas.

XIAO WANG was born in Wuyi, China in 1977. she received master's degree in mechanical and electronic engineering from Zhejiang University of technology. she is a lecturer in Xingzhi College Zhejiang Normal University and has published more than ten papers. her research interests include intelligent computing and computer aided design.

## Author Contributions

**Project administration:** Qihang Zhou.

**Validation:** Changjun Zhou, Xiao Wang.

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
