## [Decision Letter · Decision Letter 0]

23 Jul 2021

PONE-D-21-21056

Stock Prediction Based on Bidirectional Gated Recurrent Unit with Convolutional Neural Network and Feature Selection

PLOS ONE

Dear Dr. Zhou,

Thank you for submitting your manuscript to PLOS ONE. After careful consideration, we feel that it has merit but does not fully meet PLOS ONE’s publication criteria as it currently stands. Therefore, we invite you to submit a revised version of the manuscript that addresses the points raised during the review process.

We look forward to receiving your revised manuscript.

Kind regards,

Tao Song

Academic Editor

PLOS ONE

Journal Requirements:

2. Please remove your figures from within your manuscript file, leaving only the individual TIFF/EPS image files, uploaded separately.  These will be automatically included in the reviewers’ PDF.

Reviewers' comments:

Reviewer's Responses to Questions

**Comments to the Author**

1. Is the manuscript technically sound, and do the data support the conclusions?

Reviewer #1: Yes

Reviewer #2: Partly

2. Has the statistical analysis been performed appropriately and rigorously? 

Reviewer #1: Yes

Reviewer #2: No

3. Have the authors made all data underlying the findings in their manuscript fully available?

Reviewer #1: Yes

Reviewer #2: No

4. Is the manuscript presented in an intelligible fashion and written in standard English?

Reviewer #1: Yes

Reviewer #2: No

5. Review Comments to the Author

Reviewer #1: The questions raised in the introduction are properly answered in the conclusions. The results are efficient and promising.

It would be better if more literature could cited to support the relevant studies. It is suggested to cite the following literature to enrich the related work.

Song, T.; Wang, Z.; Xie, P.; A novel dual path gated recurrent unit model for sea surface salinity prediction. Journal of At-mospheric and Oceanic Technology, 2020, 37, 317-325.

Pang S, Xie P, Xu D, Meng F, Tao X, Li B, Li Y, Song T. NDFTC: A New Detection Framework of Tropical Cyclones from Meteorological Satellite Images with Deep Transfer Learning. Remote Sensing. 2021; 13(9):1860. https://doi.org/10.3390/rs13091860

Meng F, Song T, Xu D, et al. Forecasting tropical cyclones wave height using bidirectional gated recurrent unit[J]. Ocean Engineering, 2021: 108795.

Song, T.; Jiang, J.; Li, W. A deep learning method with merged LSTM Neural Networks for SSHA Prediction. IEEE Journal of Selected Topics in Applied Earth Observations and Remote Sensing, 2020, 13, 2853-2860.

Also, the font colour in Figure 2 is too light and it is recommended that it be changed to a darker colour.

Besides, in the acknowledgements section, it is suggested that the specific work of the individual authors be listed.

Reviewer #2: 1、Many words in the text have “-“ in the middle, such as “variable” in line 31, line 36 “effectively”, line 66 “increase”, line 82 “follow”, etc. In line 43, the first letter of “The” should not be capitalized.

Abstract：

2、The second sentence in the abstract wants to express the advantages of deep learning over traditional machine learning, so the comparative level is more appropriate here. "Achieve good fitting effect" is suggested to be changed to "achieve better fitting effect".

3、In the final description of the abstract, you can use refined language to add some information, such as which single comparison models were used, and the specific results show that the mixed model is better.

Research Methodology- Feature Selection (FS)- Filter Selection：

4、The explanation of formula (1) in Filter Selection is not clear enough. Only the meaning of x_i^j is explained in the article, and the other variables are not explained.

Method Based on FS-CNN-BGRU：

5、 The FS-CNN-BGRU method introduced in lines 177-189 only introduces the general process, but the organization and description of the steps are not perfect. It is recommended that you start with a subtitle for each step, and then introduce this step, and the format of the subtitle is as uniform as possible.

The fourth step of the general process is the establishment of the FS-CNN-BGRU model. Although the specific process has been described in the text above, a simple summary here will make the description of the steps more complete.

Experiment- The Construction of Model：

6、The segment of line 252 is too long. It is recommended to segment after “In” at the end of the line.

Experiment- Evaluating Indicator：

7、Why choose two evaluation indicators: MAPE and R^2？Which aspects of the model are evaluated separately by these two evaluation indicators?

Experimental Result：

8、There are no small headings in the experimental results. In order to make the structure clearer, the part should be divided into small headings according to different experimental settings.

9、Figure 5 ~ 12 did not find the corresponding explanation or reference in the text.

10、The optimal setting of the number of GRU units in the hybrid model is determined according to LSTM model. Why not directly use GRU to experiment and determine the optimal number of units?

11、The time step of the experiment is set to 1 day for forecast of 50 days. How is the time step of 50 days determined? If other step size FS-CNN-BGRU models are used, can they still achieve the best results?

6. PLOS authors have the option to publish the peer review history of their article (what does this mean?). If published, this will include your full peer review and any attached files.

Reviewer #1: No

Reviewer #2: No

---

## [Author Response · Author response to Decision Letter 0]

31 Jul 2021

Dear editor and reviewers:

Thank you for your letter and the reviewers’ comments on our manuscript entitled" Stock prediction based on bidirectional gated recurrent unit with convolutional neural network and feature selection". Your comments are of great help and guiding significance for improving the quality of this paper, improving the author's writing level and future scientific research. We have carefully studied the reviewers' comments and made corresponding amendments. Meanwhile, we apologize to the reviewers for the trouble caused by this manuscript. The reply to the comments is as follows (the reply is shown in blue).

Replies to the reviewers’ comments:

Reviewer #1:

1). The questions raised in the introduction are properly answered in the conclusions. The results are efficient and promising. It would be better if more literature could cited to support the relevant studies. It is suggested to cite the following literature to enrich the related work.

Response: Thank you for your valuable suggestions on revising this manuscript. Now we have added the four references you mentioned to the manuscript to enrich the relevant work. See [5], [31], [33], [38] for details.

2). Also, the font colour in Figure 2 is too light and it is recommended that it be changed to a darker colour.

Response: Responding to your request, we have changed the font color to a deeper color for viewing.

3). Besides, in the acknowledgements section, it is suggested that the specific work of the individual authors be listed.

Response: Your suggestion is very useful to us. We have added the relevant information of the three authors mentioned in the manuscript to the acknowledgements section.

Reviewer #2:

1). Many words in the text have “-“ in the middle, such as “variable” in line 31, line 36 “effectively”, line 66 “increase”, line 82 “follow”, etc. In line 43, the first letter of “The” should not be capitalized.

Response: Thank you very much for pointing out the problems for this manuscript. This is our negligence, and we apologize. Now we have examined the whole manuscript and removed irrelevant information.

2). The second sentence in the abstract wants to express the advantages of deep learning over traditional machine learning, so the comparative level is more appropriate here. "Achieve good fitting effect" is suggested to be changed to "achieve better fitting effect".

Response: Thank you for your suggestion. We have changed "achieve good fitting effect" to "achieve better fitting effect".

3). In the final description of the abstract, you can use refined language to add some information, such as which single comparison models were used, and the specific results show that the mixed model is better.

Response: Your modification comments are very valuable to us, Thank you very much. We added the following information to the final description of the summary: In the experiment, we used single CNN, LSTM and GRU models and mixed models CNN-LSTM, CNN-GRU and FS-CNN-BGRU (the model used in this manuscript).

4). The explanation of formula (1) in Filter Selection is not clear enough. Only the meaning of x_i^j is explained in the article, and the other variables are not explained.

Response: The lack of other relevant information about Formula 1 is our problem. So we add the following statement to explain other variables in the manuscript: (1) The construction method is as follows: Selecting the nearest neighbor $x_{i, n h}$ from the similar samples of $x_i$ , which is called guessing the nearest neighbor. Selecting a nearest neighbor $x_{i, n m}$ in heterogeneous samples is called guessing the wrong nearest neighbor. (2) It can be seen from the above formula that for the value corresponding to a feature, the closer the same kind is, the farther the different kind is, and the larger corresponding statistics will be.

5). The FS-CNN-BGRU method introduced in lines 177-189 only introduces the general process, but the organization and description of the steps are not perfect. It is recommended that you start with a subtitle for each step, and then introduce this step, and the format of the subtitle is as uniform as possible. The fourth step of the general process is the establishment of the FS-CNN-BGRU model. Although the specific process has been described in the text above, a simple summary here will make the description of the steps more complete.

Response: Thank you for your valuable comments. Now we have deleted the contents of lines 177-189 and put them into the "Experiment".

6). The segment of line 252 is too long. It is recommended to segment after “In” at the end of the line.

Response: Responding to your request, we have listed "In" and the content after "In" as a separate paragraph.

7). Why choose two evaluation indicators: MAPE and R^2？Which aspects of the model are evaluated separately by these two evaluation indicators?

Response: We are very sorry for your questions about this manuscript. We have made the following explanations: (1) during the experiment, if other evaluation indexes are used, the following problems may occur: if the stock price of the selected stock is not high, the accuracy of the model is not high, but the results calculated by using other evaluation indexes may be very good. On the contrary, if the stock is very high, the accuracy of the model is also very high, but the calculated results are not good, In other words, the quality of the model completely depends on the stock price, which is unacceptable. Therefore, an objective evaluation index should be selected to measure the quality of the model, so MAPE is selected. However, a single MAPE can not well evaluate the quality of the model, so R^2 is added. (2) The full name of MAPE is Mean Absolute Error. MAPE evaluates the accuracy of the model. 0% is the perfect model and 100% is the inferior model. The closer it is to 0%, the better. R ^ 2 also evaluates the accuracy of the model. 0 is the inferior model and 1 is the perfect model. The closer it is to 1, the better. These two evaluation indicators can accurately evaluate the advantages and disadvantages of a model.

8). There are no small headings in the experimental results. In order to make the structure clearer, the part should be divided into small headings according to different experimental settings.

Response: Your suggestions are very helpful to improve the quality of our manuscript. Thank you for your suggestions. We have added the following small headings according to your requirements: Determine the number of convolution kernels, Determine the number of lstm units and Comparison of results of various models.

9). Figure 5 ~ 12 did not find the corresponding explanation or reference in the text.

Response: This is our negligence. We apologize again. Now we have put "The forecast chart (real price and prediction price) of the eight stocks are shown in Figure 5 to 12" in lines 363-364 of the manuscript.

10). The optimal setting of the number of GRU units in the hybrid model is determined according to LSTM model. Why not directly use GRU to experiment and determine the optimal number of units?

Response: Thank you for your question. Our answers to your question are as follows: because the model used in the comparative paper [14] is LSTM and GRU is a variant of LSTM, so the LSTM model is used to determine the best setting of the number of GRU units in the hybrid model.

11). The time step of the experiment is set to 1 day for forecast of 50 days. How is the time step of 50 days determined? If other step size FS-CNN-BGRU models are used, can they still achieve the best results?

Response: We are very sorry for your confusion in this regard. We select the data from January 1, 1991 to December 31, 2020 as the data input (assuming 2000 days of data), and then randomly select a time step of 100, that is, use the data from the first day to the 100th day to predict the data of 101 days, and so on to get the data of 2000 days - 100 days = 1900 days. Then, put the data of 1900 days into the model for prediction and compare the results. On this basis, we also use other time steps to predict the stock price. Other time steps sometimes perform well, but in general, for the eight stocks in the manuscript, 50 days is the best time step.

Kind regards

Qihang Zhou

Email: 825605142@qq.com

Xiao Wang

Email: tianzhu213@zjnu.cn

Corresponding author: Changjun Zhou

E-mail address: zhou-chang231@163.com

---

## [Decision Letter · Decision Letter 1]

27 Dec 2021

Stock prediction based on bidirectional gated recurrent unit with convolutional neural network and feature selection

PONE-D-21-21056R1

Dear Dr. Zhou,

We’re pleased to inform you that your manuscript has been judged scientifically suitable for publication and will be formally accepted for publication once it meets all outstanding technical requirements.

Kind regards,

Tao Song

Academic Editor

PLOS ONE

Additional Editor Comments (optional):

Reviewers' comments:

Reviewer's Responses to Questions

**Comments to the Author**

1. If the authors have adequately addressed your comments raised in a previous round of review and you feel that this manuscript is now acceptable for publication, you may indicate that here to bypass the “Comments to the Author” section, enter your conflict of interest statement in the “Confidential to Editor” section, and submit your "Accept" recommendation.

Reviewer #1: All comments have been addressed

Reviewer #2: All comments have been addressed

2. Is the manuscript technically sound, and do the data support the conclusions?

Reviewer #1: Yes

Reviewer #2: Partly

3. Has the statistical analysis been performed appropriately and rigorously? 

Reviewer #1: Yes

Reviewer #2: Yes

4. Have the authors made all data underlying the findings in their manuscript fully available?

Reviewer #1: Yes

Reviewer #2: Yes

5. Is the manuscript presented in an intelligible fashion and written in standard English?

Reviewer #1: Yes

Reviewer #2: Yes

6. Review Comments to the Author

Reviewer #1: Every suggestion is taken seriously, which is a remarkable and admirable attitude.

The Submission has been greatly improved and is worthy of publication.

My advice is to accept this article as soon as possible. This paper will play an active and important role in the research of related fields.

Reviewer #2: The author answered all my questions very specifically and I have no further comments, so I guess it's acceptable.

7. PLOS authors have the option to publish the peer review history of their article (what does this mean?). If published, this will include your full peer review and any attached files.

Reviewer #1: No

Reviewer #2: No

---

## [Editor Report · Acceptance letter]

27 Jan 2022

PONE-D-21-21056R1 

Stock prediction based on bidirectional gated recurrent unit with convolutional neural network and feature selection 

Dear Dr. Zhou:

I'm pleased to inform you that your manuscript has been deemed suitable for publication in PLOS ONE. Congratulations! Your manuscript is now with our production department. 

Kind regards, 

on behalf of

Dr. Tao Song 

Academic Editor

PLOS ONE